# Zygomatic Implants Placed in Immediate Function through Extra-Maxillary Surgical Technique and 45 to 60 Degrees Angulated Abutments for Full-Arch Rehabilitation of Extremely Atrophic Maxillae: Short-Term Outcome of a Retrospective Cohort

**DOI:** 10.3390/jcm10163600

**Published:** 2021-08-16

**Authors:** Armando Lopes, Miguel de Araújo Nobre, Ana Ferro, Carlos Moura Guedes, Ricardo Almeida, Mariana Nunes

**Affiliations:** 1Oral Surgery Department, Maló Clinic, Avenida dos Combatentes, 43, Piso 9, 1600-042 Lisboa, Portugal; alopes@maloclinics.com (A.L.); aferro@maloclinics.com (A.F.); mnunes@maloclinics.com (M.N.); 2Research, Development and Education Department, Maló Clinic, Avenida dos Combatentes, 43, Piso 11, 1600-042 Lisboa, Portugal; 3Prosthodontics Department, Maló Clinic, Avenida dos Combatentes, 43, Piso 10, 1600-042 Lisboa, Portugal; cguedes@maloclinics.com (C.M.G.); ralmeida@maloclinics.com (R.A.)

**Keywords:** zygomatic implants, immediate loading, angulated abutments, atrophic jaw, maxilla

## Abstract

The use of new devices for the rehabilitation of the severely atrophic maxillae needs validation. We aimed to report the short-term outcome of severely atrophic jaws rehabilitated with zygomatic implants with no implant head angulation placed extramaxillary in conjunction with standard implants. Forty-four patients were consecutively included with 77 zygomatic implants (31 abutments of 45 degrees and 46 abutments of 60 degrees) and 115 standard implants. Outcome measures were prosthetic survival, implant/abutment success, complications, modified plaque index (mPLI), modified bleeding index (mBI), mucosal seal efficacy evaluation (MSEE) >4 mm, and Zygomatic implants classification level (ZICL). Two patients (4.5%) were lost to follow-up. No prosthesis was lost; one patient lost one zygomatic implant; two angulated abutments of 60 degrees needed to be replaced in one patient due to an aesthetic complaint; rendering a cumulative success rate at 2-years of 95.3% and 95.9% using patient and implant/abutment as unit of analysis, respectively. Mechanical and biological complications occurred in 13 and six patients, respectively; all resolved. The median mPLI and mBI was 1; MSEE > 4 mm occurred in 17% and 21% of patients at 1- and 2-years, respectively; ZICL1 was registered in 80% of patients. The current protocol enabled good short-term outcomes.

## 1. Introduction

Zygomatic implants are one of the alternatives to bone grafting for fixed prosthetic rehabilitation in the absence of enough residual bone, based on the premises of decreasing treatment time, reducing the number of surgeries and anaesthetic procedures, eliminating donor graft site morbidity, and reducing the overall cost of surgical and prosthetic treatment while maintaining excellent patient satisfaction outcomes [1,2].

The indications of zygomatic implants range from the treatment of atrophic maxilla [3,4], ectodermal dysplasia [5] to maxillary reconstruction after maxillectomy in cancer patients [6,7] with favorable results considering cumulative survival rates (CSRs) ranging between 95.2% and 98.6% with 2- to 12-years follow-up [1,8,9,10,11]. Moreover, considering immediate function, despite the scarce number of studies, CSRs of up to 98.33% with follow-ups up to 7-years were reported [9,11]. Nevertheless, complications associated with zygomatic implant treatment include sinusitis, soft tissue infection, paraesthesia, orbital perforation, and oroantral fistulas [1,12], with sinusitis as a particularly difficult complication to manage [9]. Further disadvantages may be associated with the insertion technique: the internal technique [7], considering the insertion of the zygomatic implant intra-sinus, with a potential increased probability of sinus complications and a bulky prosthesis caused by the palatal emergence. The extra-maxillary surgical technique aims to overcome these limitations, by placing the zygomatic implant extra-maxillary (external to the maxillary sinus before anchoring in the zygomatic bone, covered only by soft tissue along its lateral maxillary surface) [11] providing the preservation of the Schneiderian membrane and a decreased vestibular-palatine width of the prosthesis due to the more coronal emergence of the zygomatic implant [13]. Nevertheless, in the presence of extreme angulations, material alternatives to overcome that limitation are lacking. In this sense, the inclusion of 45- and 60-degrees abutments could benefit the rehabilitations providing the necessary compensation in the degrees of angulation.

The aim of the present study was to report the short-term outcome of fixed prosthetic rehabilitations of the atrophic maxillae supported by zygomatic implants placed through the extra-maxillary surgical technique and 45- and 60-degrees angulated abutments.

## 2. Materials and Methods

This study was approved by an Ethical Committee (Ethical Committee for Health, authorization no. 003/2019). This retrospective cohort study was performed at a private rehabilitation center between July 2016 and August 2020. The study included 44 consecutively treated patients (33 women; 11 men). Patients were identified from the medical records as having consented to complete edentulous maxillary rehabilitation with the use of implants inserted into the Zygomatic bone. Inclusion criteria were candidacy for immediate fixed implant-supported rehabilitation of the complete edentulous maxilla with extreme horizontal and vertical bone loss and pneumatization of the maxillary sinuses (C-VI and D-V or D-VI; Cawood-Howell classification) [14]. There were no patients judged to have any exclusion criteria including active radiotherapy or chemotherapy or presenting emotional instability. The medical history of each patient was reviewed, and the diseases were coded using the International Classification of Diseases, version 11 (ICD-11) [15]. 

### 2.1. Surgical Protocol

The patients were rehabilitated by using one or two zygomatic implants in conjunction with conventional implants. All implants were placed in immediate function. The zygomatic implants used in this study were NobelZygoma 0° with a TiUnite surface (Nobel Biocare AB, Göteborg, Sweden), and typically emerged between the first premolar and the first molar on the residual crest of the ridge, near its center) [16]; The study abutments were 45° and 60° Multi-unit abutments External Hex RP6 mm of height (Nobel Biocare AB, Figure 1). The surgical protocol followed previous indications for zygomatic implants inserted through the extramaxillary surgical technique [17,18]. A clinical examination with a preoperative orthopantomography and a cone beam computed tomography (CBCT) scan was used to plan the surgery. In this study, conventional maxillary anchored implants (NobelSpeedy groovy, Nobel Biocare AB) were inserted in the inter-canine area; while for the posterior region, the presence of a maxillary bone quantity of D-V or D-VI (Cawood-Howell classification) [14] implied the insertion of one implant with zygomatic anchorage (All-on-4 Hybrid^TM^; Nobel Biocare AB). The surgery was performed under general anesthesia or local anesthesia, according to the patient’s wishes. A mucoperiosteal incision was made along the crest of the ridge, slightly palatal, from the molar area to the contralateral molar area, with buccal vertical releasing incisions made posteriorly to expose the zygomatico-maxillary buttress and the prominence of the zygoma. Flap reflection allowed for infra-orbital nerve identification and protection as well as direct observation of the lateral aspect of the zygomatic bone. 

Zygomatic implant lengths and positions were determined peri-operatively and were dependent on the anatomy of the region. The “channel” osteotomy began as distal as possible at the maxillary crest level with a channel drill directed along a planned implant direction which maintained a minimum safe distance of approximately 3 mm from the posterior-inferior edge of the zygomatic bone, attempting not damage the sinus membrane. The sinus membrane was then carefully elevated from the internal wall of the sinus. This “channel” facilitated access and an optimal path to the Zygomatic bone for the implant drills without any tissue interference, and typically helped to “buttress” the implant against the lateral maxillary wall. The zygomatic implants inserted through extra-maxillary technique were placed with an insertion torque of at least 30 N-cm for sufficient primary stability. This protocol allowed to position the implant’s head near the buccal aspect of the residual crest (less palatal, compared with the surgical protocol by Brånemark et al. 2004) [8]. The Multi-unit abutments (Nobel Biocare AB) of 45- and 60-degrees and 6 mm of height were connected to the NobelZygoma 0° implants (Nobel Biocare) adjusting the mesial tilting of the implants and allowing the prosthetic screw access to be positioned on the occlusal aspect of the prosthetic teeth. Straight multi-unit abutments (Nobel Biocare, AB) were connected to the standard anterior implants. In some situations, to reposition the insertion axis enabling a parallel position between all the implants, the straight abutments were replaced by 30-degrees angulated abutments. The edges of the flaps were re-approximated tension free with interrupted sutures. Buccal keratinized gingiva was preserved, especially around the implants.

### 2.2. Immediate and Final Prosthetic Protocol

The immediate and final prosthetic protocols consist of standardized procedures reported previously [19]. Concerning the immediate prosthesis, a high-density acrylic resin (PalaXpress Ultra, Heraeus Kulzer GmbH, Hanau, Germany) prosthesis with Temporary Coping Multi-Unit Titanium (Nobel Biocare AB) was manufactured at the dental laboratory and inserted the same day. 

Typically, six months after surgery, according to patient preference and clinical considerations, a definitive restoration was connected: a titanium framework (NobelProcera, Nobel Biocare AB) and either all-ceramic crowns (e.Max Press, Ivoclar Vivadent AG, Schaan, Liechtenstein) or acrylic resin prosthetic teeth (Heraeus Kulzer GmbH, Hanau, Germany) were used to replace the provisional prosthesis. A representative clinical case is displayed in Figure 2.

### 2.3. Outcome Measures

Outcome measures were evaluated at implant surgery and at 2 years post-surgery. The primary outcome measures were prosthetic success, implant success, abutment success and complications.
Prosthetic success was judged in terms of function, being considered a failure if needed to be replaced by a new prosthesis.Implants were considered a success considering [17]: (1) it fulfilled its purported function as support for reconstruction; (2) it was stable when individually and manually tested [19]; (3) no signs of persistent prevalent infection observed; (4) demonstrated a good aesthetic and functional outcome of the rehabilitation; and (5) allowed fabrication of the implant-supported fixed prosthesis which provided patient comfort and hygiene. In the situations where the implants did not fulfil the criteria for success but remained in site, these were considered survivals. In situations of implant removal, these were considered as failures.Abutments were considered a success considering: the fulfilment of their purported function as support for the reconstruction; absence of fractures; absence of aesthetic or functional complaints from the patient.Complication parameters assessed were: fracture or loosening of mechanical and prosthetic components (mechanical complications); soft tissue inflammation, fistula formation, pain, or maxillary sinus infections, peri-implant pathology (probing pocket depths >4 mm together with bleeding of the peri-implant soft tissue and/or presence of dental plaque) (biologic complications); aesthetic complaints of the patient or dentist (aesthetic complications); phonetic complaints, masticatory complaints, comfort complaints or hygienic complaints (functional complications).

Secondary outcome measures were the modified plaque index (mPLI), modified bleeding index (mBI), mucosal seal efficacy evaluation (MSEE) and the Zygomatic implants clinical level (ZICL).
Modified plaque index (mPLI) recorded in an ordinal scale between 0 and 3 (0: no plaque visible; 1: plaque only visible after the insertion of the probe; 2: plaque visible with the naked eye; and 3: abundance of soft matter) [20];Modified bleeding index (mBI), recorded in an ordinal scale between 0 and 3 (0: no bleeding visible; 1: isolated bleeding spots visible; 2: bleeding forms a confluent red line on the margin; and 3: heavy or profuse bleeding) [20];Mucosal seal efficacy evaluation (MSEE) was assessed by inserting a 0.25 Ncm calibrated plastic periodontal probe (Hawe-Neos, Bioggio, Switzerland) in the sulcus of the zygomatic implant until a maximum depth of 4 mm and recorded as “0“ if the probe stopped before 4 mm of depth or as “1” if the probe did not stop before 4 mm of depth [21];Zygomatic Implants Clinical Level (ZICL) was computed considering the MSEE, mPLI and mBI clinical indexes: 21 ZICL 1 (MSEE = 0; mPLI = 0; mBI = 0), ZICL2 (MSEE = 1; mPLI = 0; mBI = 0), ZICL3 (MSEE = 1; mPLI = 0; mBI = 1–3), ZICL4 (MSEE = 1; mPLI = 1–3; mBI = 0–3).

### 2.4. Statistical Evaluation

Descriptive statistics were applied to the variables of interest (complications, mPLI, mBI, MSEE, ZICL). The cumulative survival and success rates were estimated using the Kaplan-Meyer product limit estimator when using the patient as unit of analysis (first implant failure in any given patient). The cumulative survival and success rates were estimated using life tables when using the implant as unit of analysis.

## 3. Results

### 3.1. Sample

The study included 44 consecutively treated patients (33 women and 11 men), with an age range of 27–72 years (mean = 54.4 years) followed for 2 years. A total of 23 patients had at least one systemic condition according to the ICD-11, with 11 patients who were smokers, and 16 patients (two of the patients who were smokers) presented with the following conditions: hepatitis (n = 1), cardiovascular disease (n = 17), endocrine dysfunction (n = 4), diabetes (n = 3), digestive (n = 1), oncologic condition (n = 3), depressive disorder (n = 2), disease of respiratory system (n = 2), autoimmune condition (n = 1). Eight patients presented more than one condition. Sixteen patients were diagnosed as heavy bruxers prior to the prosthetic rehabilitation. Two patients (female patients with 64 and 73 years of age representing 4.5% of the sample) with four zygomatic implants and four study abutments were lost to follow-up during the first year becoming unreachable.

Seventy-seven extra-maxillary zygomatic implants were inserted with connection of multi-unit angulated abutments of 60- (n = 46) and 45-degrees (n 31) and 6 mm of height; together with the insertion of 115 standard implants (Table 1).

### 3.2. Prosthetic Success

A total of 44 completely edentulous maxillary rehabilitations were performed. Despite the occurrence of implant failures, the prostheses were adapted to the replaced dental implants allowing it to remain in function, giving a prosthetic survival rate of 100%.

### 3.3. Implant Survival

One extra-maxillary zygomatic implant failed to integrate in one patient and was lost after 7 months, rendering a 97.7% and 98.7% survival rate at 2 years using the patient and implant as unit of analysis, respectively (Table 2). The prosthesis remained in function supported by three implants for five months, and after that period another zygomatic implant was inserted and not loaded. The patient was lost to follow-up afterwards becoming unreachable. Three patients lost five standard implants rendering a 95.6% CSR at two years. 

No study abutments fractured, thus achieving a survival rate of 100%. A total of five study abutments of 45/60 degrees (6.5%) were replaced during the manufacture of the definitive prosthesis for adaptation to the new prostheses (Table 3).

### 3.4. Complications

Mechanical complications occurred in 13 patients (29.6%) (Table 3). The situations were resolved in all patients by repairing the prosthesis (fractures), tightening the prosthetic components (abutment screw loosening), adjusting the occlusion and manufacturing night-guards. No functional complications were registered. Aesthetic complications were registered in one patient (2.8%) that led to the replacement of the study abutments (previously described in Table 3). Biological complications occurred in 6 patients (13.6%) and 6 implants (7.8%), with all situations resolved (Table 4). 

### 3.5. Clinical Evaluation Parameters

The mPLI and mBI registered a median of 1 at both 1- and 2-years of follow-up (Figure 3 and Figure 4). The incidence of MSEE > 4 mm was 17% and 21% at 1- and 2-years of follow-up, respectively. Most zygomatic implants were classified as ZICL 1 at 1- and 2-years of follow-up (Figure 5).

### 3.6. Study Zygomatic Implant and Study Abutment Success

Considering the implant failure from one patient at seven months and the aesthetic complaint from a second patient at 22 months, the cumulative success rate at two years for the study zygomatic implants was 95.3% and 95.9% using the patient and implant as unit of analysis; with equal figures also for the study abutments (Table 5). 

## 4. Discussion

The present study reported the short-term outcome of fixed prosthesis supported by immediate function zygomatic implants inserted extra-maxillary with 45- and 60-degrees angulated abutments in conjunction with standard implants for the rehabilitation of the severely atrophic maxillae, with a high survival and success rates for prostheses, implants, and abutments. The implant survival of 98.7% is comparable to what is reported in the literature for zygomatic implants inserted either through classical or extra-maxillary surgical techniques. Goiato et al. [10] in a systematic review evaluating 25 clinical studies on implants inserted in the zygomatic bone for maxillary rehabilitation reported 97.86% of implant CSR at the 2–3 years follow-up time. Concerning the extra-maxillary surgical technique, 99% to 100% implant CSR were registered on the same follow-up time [18,21].

Most study abutments were functional during the follow-up up of the study. However, a total of three study abutments were replaced for technical reasons. The replacement of abutments occurred either due to modification deemed necessary between the provisional and definitive prostheses (not related to patient complaints) or due to aesthetic complaints (from one patient) that was dissatisfied with visible abutments on the posterior segment of the full-arch prosthesis. Nevertheless, both abutments were functioning correctly and located above the smile line without any aesthetic compromise. It has been previously reported that designing a prosthetic restoration in these patients can be a prosthodontic and laboratorial challenge considering the anatomic limitations that affect implant placement, namely the fact that if zygomatic implants are deemed necessary then it is due to lack of bone [22]. Nevertheless, the complication was resolved by replacing the study abutments by 30 degrees abutments.

The incidence of mechanical complications was high (around 30%), a situation that is always burdensome for both patients and clinicians due to the necessity of further interventions to resolve the complication. However, the mechanical complications occurred primarily on the provisional prostheses and in a specific set of patients, all with implant-supported prostheses as opposing dentitions and four of the patients with further bruxing habits, two common risk indicators for mechanical complications previously reported in other investigations both on zygomatic implants [11] and standard implants [23,24,25,26]. The authors propose a prosthetic and maintenance protocol with short intervals and regular assessment of clinical, oral hygiene and occlusion parameters to maximize the probability of success.

Biological complications occurred in 13.6% of patients during the follow-up of the study. This result is comparable to a previously reported study for extra-maxillary zygomatic implants with a similar follow-up (12.8%) [18]. The biological complications were all resolved through non-surgical interventions except for one implant in one patient that required a surgical intervention and antibiotic therapy to resolve the occurrence of an abscess at 8 months of follow-up. Nevertheless, there were no incidences of sinusitis reported in the present study which is one of the aims of the extra-maxillary surgical protocol with reduced manipulation of the sinus membrane, and consequently lower risk of sinusitis.

Bacterial plaque registered in the present study was of minor accumulation in most patients, a result that may be related to the extra-maxillary technique. This technique implies a more crestal emergence position of the zygomatic implants compared to the classical technique and therefore providing a thinner prosthesis, allowing the patient to have better access for self-care [13]. The low bleeding levels find parallel in the literature where previous publications reporting the insertion of zygomatic implants through the same surgical approach registered identical mBI values at 1-, 2- and 3-years of follow-up [18]. However, the presence of bleeding can be the result of an inflammatory response to bacterial plaque accumulation as previously reported [27]. 

It is known that zygomatic implants inserted intra-sinus and with maxillary anchorage have deeper pockets compared to standard implants [28,29]. This result was further confirmed in previous publications concerning extra-maxillary zygomatic implants, where a 9.8% rate of implants with pockets > 4 mm (n = nine implants) was reported at 2 years [18], and a 9% increase when compared to standard implants [21]. This tendency was attributed to the surgical protocol, in which the zygomatic implants were inserted extra-maxillary, with only zygomatic anchorage and therefore only soft tissue coverage in their coronal third. 

The ZICL evaluation refers to an index for evaluating the prognosis of a zygomatic implants inserted through the extra-maxillary surgical technique, progressing with a crescent probability of a negative outcome from levels 1 to 4 [21]. In the present report, 80% of patients were classified as ZICL1 and 20% as ZICL4, results that are comparable to a previous publication [21], (with 70% and 23%, respectively) and implies a good prognosis for most patients.

The results of the present study should be interpreted with caution considering its limitations that include the retrospective design and the short-term follow-up. Nevertheless, the low percentage of patients lost to follow-up enabled to evaluate a significant portion of the sample (94.5%) allowing to answer the research questions of the present study. The generalizability of the study results is limited to patients with extremely atrophied maxillae rehabilitated through zygomatic implants.

Future research should focus on the evaluation of longer outcomes for these patients with severely atrophic maxillae rehabilitations, including long term mechanical, soft tissue, and quality of life assessments.

## Figures and Tables

**Figure 1 jcm-10-03600-f001:**
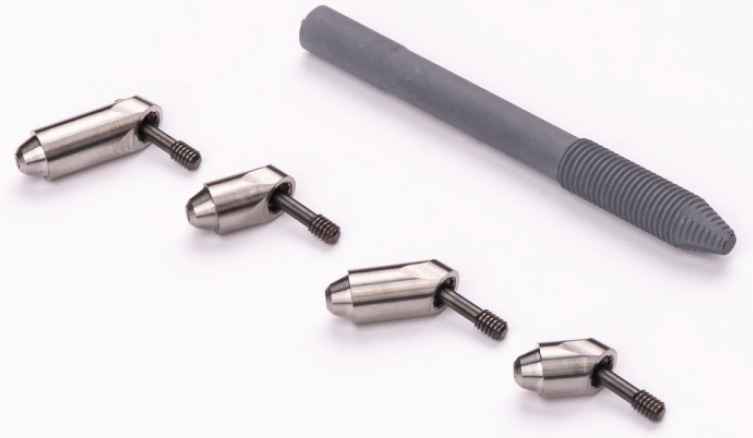
Study implant (NobelZygoma 0°; Nobel Biocare) and study abutment (Multi-unit abutments of 45 and 60 degrees and 6 mm of height; Nobel Biocare).

**Figure 2 jcm-10-03600-f002:**
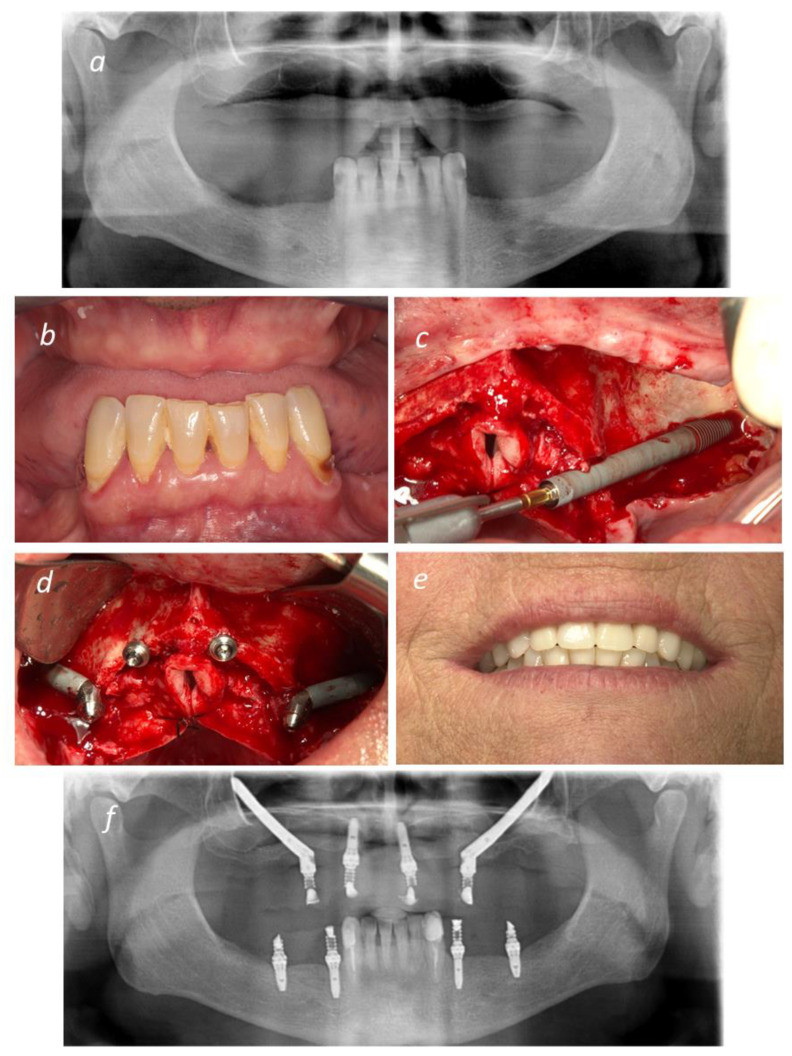
Representative figures of an All-on-4 Hybrid rehabilitation: (**a**) Pre-operative Orthopantomography; (**b**) Pre-operative intra-oral photograph in frontal view; (**c**) Insertion of NobelZygoma 0° implant through the Extramaxillary surgical technique; (**d**) Two NobelZygoma 0° implants inserted bilaterally in the posterior region with 45° Multi-unit abutments and two anterior NobelSpeedy Groovy implants with straight Multiunit abutments (All-on-4 Hybrid); (**e**) Immediate Provisional prosthesis loaded on the day of surgery; (**f**) Post-operative Orthopantomography.

**Figure 3 jcm-10-03600-f003:**
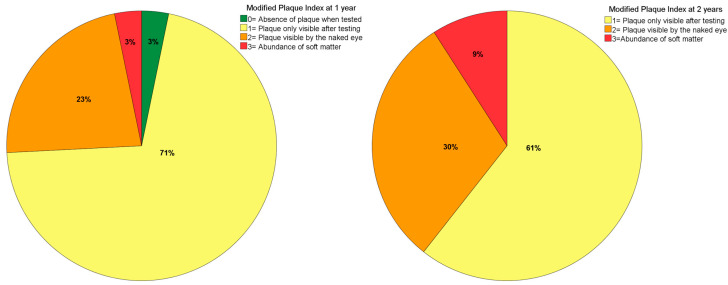
Plaque index levels at 1- and 2-years of follow-up.

**Figure 4 jcm-10-03600-f004:**
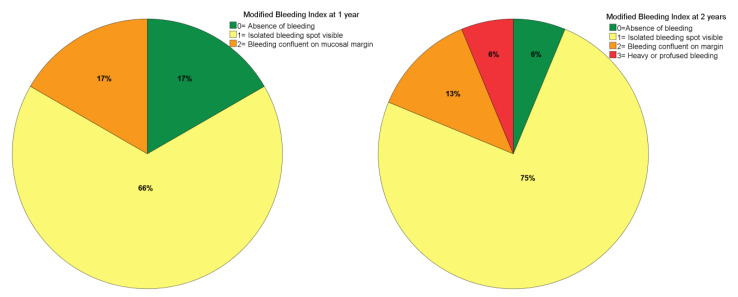
Modified bleeding index levels at 1- and 2-years of follow-up.

**Figure 5 jcm-10-03600-f005:**
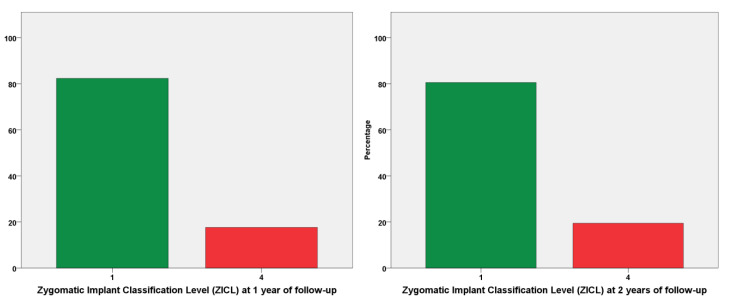
Zygomatic implant classification levels (ZICL) at 1- and 2-years of follow-up. The ZICL 1 level is considered the better prognostic considering the preventive aspect of a biological complication, with absence of MSEE > 4 mm, while a ZICL 4 is considered the worst prognostic level from a preventive point of view, with the presence of MSEE > 4 mm and simultaneous presence of plaque and bleeding. The sample was distributed along the two ZICL levels, with an increase of 1% in zygomatic implants with ZICL 4 level from the first to the second year of follow-up (18% to 19%).

**Table 1 jcm-10-03600-t001:** Study population: Implant type, position of emergence and loading regimen.

Patients	Age	Sex	Location of Implant Emergence									
			Right (1st Quadrant)					Left (2nd Quadrant)				
			First Molar	Second Premolar	First Premolar	Canine	Lateral Incisor	Lateral Incisor	Canine	First Premolar	Second Premolar	First Molar
1	62	M		Z 5 × 40 ^◊^		S 5 × 8.5			S 5 × 8.5 ^F^	S 5 × 10 ^F^		S × 10 **
2	50	F		Z 5 × 45 ^●^			S 3.3 × 13	S 3.3 × 15			Z 5 × 45 ^●^	
3	52	F			Z 5 × 42.5 ^◊^		S 4 × 10	S 4 × 10		Z 5 × 37.5 ^◊^		
4	45	M			Z 5 × 40 ^●^		S 4 × 11.5	S 4 × 11.5			Z 5 × 40 ^●^	
5	62	F		Z 5 × 40 ^◊^		S 4 × 7	S 4 × 7	S 4 × 7 **			Z 5 × 40 ^◊^	
6	54	F		S 4 × 11.5			S 3.3 × 11.5	S 3.3 × 11.5			Z 5 × 42.5 ^◊^	
7	47	F		Z 5 × 45 ^◊^		S 4 × 11.5	S 4 × 8.5	S 4 × 8.5	S 3.3 × 11.5		Z 5 × 42.5 ^◊^	
8	51	F		Z 5 × 40 ^◊^			S 4 × 10	S 4 × 10			Z 5 × 40 ^◊^	
9	61	F		Z 5 × 42.5 ^●^			S 3.3 × 10	S 4 × 8.5			Z 5 × 42.5 ^●^	
10	41	F				Z 5 × 40 ^●^	S 4 × 8.5	S 4 × 8.5	Z 5 × 40 ^●^			
11	53	F			Z 5 × 45 ^◊^		S 4 × 10	S 4 × 10 ^F^		S 5 × 11.5		
12	64	F		Z 5 × 50 ^●^			S3.3 × 13	S 3.3 × 13		Z 5 × 50 ^●^		
13	69	F		Z 5 × 42.5 ^◊^			S 4 × 11.5	S 4 × 11.5			Z 5 × 42.5 ^◊^	
14	72	F		Z 4 × 45 ^^◊^ F^			S 4 × 10	S 4 × 8.5			Z 5 × 47.5 ^●^	
15	27	F		Z 5 × 35 ^◊^		S 4 × 10		S 4 × 11.5			S 4 × 13	
16	68	F		Z 5 × 42.5 ^●^		S 4 × 8.5	S 4 × 7 **	S 4 × 7	S 4 × 7		Z 5 × 42.5 ^●^	
17	53	F		Z 5 × 40 ^◊^		S 3.3 × 10	S 4 × 7	S 4 × 8.5	S 3.3 × 10		Z 5 × 40 ^◊^	
18	42	M		S 4 × 11.5			S 4 × 11.5	S 4 × 11.5			Z 5 × 45 ^●^	
19	42	F		S 4 × 10			S 3.3 × 10	S 3.3 × 10			Z 5 × 45 ^◊^	
20	47	M		Z 5 × 47.5 ^◊^		S 4 × 13		S 3.3 × 13			S 4 × 15	
21	61	F		Z 5 × 35 ^◊^			S 4 × 10	S 4 × 10			Z 5 × 40 ^◊^	
22	57	F		Z 5 × 42.5 ^●^			S 5 × 15	S 5 × 18			Z 5 × 45 ^◊^	
23	48	F		Z 5 × 45 ^●^			S 4 × 10	S 3.3 × 15	S 3.3 × 10		Z 5 × 45 ^●^	
24	45	M		Z 5 × 45 ^◊^		S 4 × 10	S 4 × 8.5	S 4 × 8.5			Z 5 × 45 ^◊^	
25	57	F		Z 5 × 45 ^◊^			S 4 × 11.5	S 4 × 11.5			Z 5 × 45 ^◊^	
26	56	M		Z 5 × 47.5 ^◊^			S 4 × 11.5	S 4 × 8.5			Z 5 × 45 ^◊^	
27	68	F		S 4 × 15			S 4 × 11.5	S 4 × 11.5			Z 5 × 45 ^◊^	
28	38	M		Z 5 × 47.5 ^●^			S 3.3 × 11.5	S 3.3 × 11.5			Z 5 × 45	
29	67	F		Z 5 × 45 ^◊^			S 4 × 13	S 4 × 13			Z 5 × 45 ^◊^	
30	48	F		Z 5 × 42.5 ^◊^		S 4 × 10 ^F^			S 4 × 10 ^F^		Z 5 × 42.5 ^◊^	
31	57	M		Z 5 × 45 ^◊^		S 4 × 15	S 4 × 7 **	S 4 × 7 **	S 4 × 13		Z 5 × 45 ^◊^	
32	59	F		Z 5 × 50 ^◊^			S 4 × 10	S 4 × 10			Z 5 × 50 ^●^	
33	49	M		Z 5 × 47.5 ^●^			S 3.3 × 10	S 4 × 10			Z 5 × 45 ^●^	
34	70	M			Z 5 × 47.5 ^◊^		S 4 × 13	S 3.3 × 13			Z 5 × 45 ^◊^	
35	55	F		Z 5 × 42.5 ^◊^			S 3.3 × 15	S 3.3 × 15			Z 5 × 42.5 ^◊^	
36	67	F		S 4 × 15			S 4 × 10	S 4 × 10			Z 5 × 40 ^●^	
37	40	F		Z 5 × 40 ^●^			S 3.3 × 10	S 4 × 10			Z 5 × 40 ^●^	
38	64	F		Z 5 × 37.5 ^●^			S 4 × 8.5	S 4 × 8.5			Z 5 × 40 ^●^	
39	63	F		Z 5 × 45 ^●^			S 3.3 × 13	S 3.3 × 13			Z 5 × 40 ^●^	
40	56	F		Z 5 × 40 ^●^			S 4 × 10	S 4 × 10			S 5 × 15	
41	56	F		Z 5 × 40 ^◊^		S 4 × 10	S 4 × 8.5	S 4 × 10	S 4 × 10		Z 5 × 42.5 ^◊^	
42	46	F		Z 5 × 35 ^●^			S 3.3 × 10	S 3.3 × 10			Z 5 × 35 ^●^	
43	64	F		Z 5 × 42.5 ^◊^		S 4 × 8.5	S 4 × 8.5	S 4 × 8.5	S 4 × 8.5		Z 5 × 42.5 ^◊^	
44	42	M			Z 5 × 45 ^◊^		S 4 × 11.5	S 4 × 11.5		Z 5 × 40 ^◊^		

M: male; F: female; ** Rescue implant not loaded; Type of implants: Z—Zygomatic extra-maxillary implant; S—Standard implant; Diameter × Length (mm); ^◊^ 60° abutment; ^●^ 45° abutment; F Implant failure.

**Table 2 jcm-10-03600-t002:** (a) Study implants and study abutments survival in complete edentulous rehabilitations using the patient as unit of analysis (Kaplan-Meyer product limit estimator). (b) Life table for cumulative survival rate of the study zygomatic implants and study abutments of 45 and 60 degrees using the implants/abutments as unit of analysis.

**(a)**						
**Time (Months)**	**Status (0 = Success; 1 = Failure *)**	**Cumulative Proportion Success at the Time**			**N of Cumulative Events**	**N of Patients at Risk**
		**Estimate**	**Std. Error**			
0	0				0	44
7	1	0.977	0.022		1	43
9	0				1	42
10	0				1	41
12	0				1	41
13	0				1	40
24	0				1	40
**(b)**						
**Duration**	**Total**	**Failed**	**Lost to Follow-Up**	**Censored**	**Survival Rate %**	**Cumulative Survival Rate %**
Placement—1 year	77	1	1	5	98.7%	98.7%
1 year—2 years	70	0	2	2	100%	98.7%
2 years—3 years	52	0	0	25	100%	98.7%

* Failure was defined as the first implant to fail in one patient.

**Table 3 jcm-10-03600-t003:** Replacement of study abutments and incidence of mechanical complications.

**Patient** **Age/Sex**	**Abutment Type**	**Follow-Up in Months**	**Reason for Change**
63/Female	45 degrees (n = 2)	4	Change of prosthetic angulation to a 60 degrees abutment from provisional to definitive prosthesis
38/Male	45 degrees	9	Change of prosthetic angulation to a 60 degrees abutment from provisional to definitive prosthesis.
68/Female	45 degrees (n = 2)	10	Change of prosthetic angulation to a 30 degrees abutment from provisional to definitive prosthesis.
55/Female	60 degrees (n = 2)	22	Change to 30 degrees abutments due to patient not being satisfied with volume and visible abutments on the posterior segment at the time of definitive prosthesis manufacture.
Mechanical complications			
**Patient** **Age/Sex**	**Condition ^a^**	**Follow-Up in Months**	**Complications**
48/Female	Heavy bruxer	1	60 degrees abutment loosening (zygomatic implant)
50/Female		3	Straight abutment loosening (standard implant)
38/Male	Heavy bruxer	4	45 degrees abutment loosening (zygomatic implant)
72/Female		6	60 degrees abutment loosening (zygomatic implant)
53/Female		7	Fracture of provisional prosthesis supported by 3 dental implants due to a standard implant failure.
68/Female		7	Line of fracture occurring at the level of implant #12 in the provisional prosthesis
46/Female		7	Fracture of provisional prosthesis occurring at the implant positions #22 to #25
64/Female	Heavy bruxer	7	60 degrees abutment loosening (zygomatic implant)
42/Male		9	Fracture of provisional prosthesis occurring at the cylinder level of implant #25
62/Male		11	Fracture of crown on position #12 in provisional prosthesis
57/Female		13	Fracture of provisional prosthesis and abutment screw loosening occurring at the level of implant #15
64/Female		17	Fracture of crown at position #12 on the metal-acrylic definitive prosthesis
49/Male	Heavy bruxer	22	Fracture of the provisional prosthesis occurring at the level of the 2 cylinders in the first quadrant.

^a^ All patients with implant-supported fixed prosthesis as opposing dentition.

**Table 4 jcm-10-03600-t004:** Biological complications occurred during the study follow-up and resolution approaches.

Patient Age/Sex	Systemic Conditions	Implant Position	Abutment Type	Follow-Up	Complication *	Resolution Approach
69/Female	Cardiovascular condition/Diabetes	25	60°	2 months	Abscess	Non-surgical ^1^ + antibiotics
41/Female	Smoker + Chronical Sinusitis	23	45°	5 months	Suppuration	Non-surgical + Antibiotics
54/Female	Absent	24	60°	8 months	Abscess	Surgical ^2^ + Antibiotics
59/Female	Smoker + Allergic Rhinitis	25	45°	11 months	Suppuration	Non-surgical + Antibiotics
56/Female	Cardiovascular condition + Endocrine condition	15	45°	11 months	Fistula	Surgical + Antibiotics
47/Male	Cardiovascular condition	15	60°	23 months	Suppuration	Non-surgical

* All complications were successfully resolved; ^1^ Non-surgical intervention comprised of curettage, polish, and irrigation with 0.2% chlorhexidine; ^2^ Surgical intervention comprised: removal of granulation tissue, decontamination of the implant surface with chlorhexidine 0.2%.

**Table 5 jcm-10-03600-t005:** (a) Study implants and study abutments success in complete edentulous rehabilitations using the patient as unit of analysis (Kaplan-Meyer product limit estimator). Life tables for cumulative success rate of the study zygomatic implants and study abutments of 45 and 60 degrees using the implants/abutments as unit of analysis.

**(a)**						
**Time (Months)**	**Status (0 = Success; 1 = Failure *)**	**Cumulative Proportion Success at the Time**			**N of Cumulative Events**	**N of Patients at Risk**
		**Estimate**	**Std. Error**			
0	0	.	.		0	44
7	1	0.972	0.022		1	43
9	0	.	.		1	42
10	0	.	.		1	41
12	0	.	.		1	41
13	0	.	.		1	40
22	1	0.953	0.033		2	39
24	0	.	.		2	39
**(b)**						
**Duration**	**Total**	**Failed**	**Lost to Follow-Up**	**Censored**	**Survival Rate %**	**Cumulative Survival Rate %**
Placement—1 year	77	1	1	5	98.7%	98.7%
1 year—2 years	70	2	2	0	97%	95.9%
2 years—3 years	66	0	0	25	100%	95.9%

* Failure was defined as the first implant to fail in one patient.

## Data Availability

The data presented in this study are available on request from the corresponding author. The data are not publicly available due to ethics.

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
