# Peer review of "Zygomatic Implants Placed in Immediate Function through Extra-Maxillary Surgical Technique and 45 to 60 Degrees Angulated Abutments for Full-Arch Rehabilitation of Extremely Atrophic Maxillae: Short-Term Outcome of a Retrospective Cohort"

_jcm, 2021, doi:10.3390/jcm10163600_

Round 1
Reviewer 1 Report
Dear Authors,
I can see that the study design is appropriate, I found it interesting. As you said in the conclusions it is important that future research focuses on evaluating long-term outcomes because it is essential to allow colleagues and patients to evaluate which is the best prosthetic rehabilitation. I think it could be accepted after a minor revision.
line 40: there is the citation [6] without the square brackets.
line 42: I suggest using the number "2" instead of "two".
line 72: you wrote "radio therapy". You should cancel the space and write "radiotherapy" (or "Radiation therapy").
line 92: from which molar to which molar the mucoperiosteal incision extends? Regardless instead of "from molar area to molar area" I suggest "from the molar area to the contralateral molar area"
line 96: in how many cases was the DTX Studio Implant Software used to plan the surgery? Why was the DTX Studio Implant Software used only in certain cases and not in others?
line 300: there is the citation [11] without the square brackets.
Author Response
Dear Authors,
1. I can see that the study design is appropriate, I found it interesting. As you said in the conclusions it is important that future research focuses on evaluating long-term outcomes because it is essential to allow colleagues and patients to evaluate which is the best prosthetic rehabilitation. I think it could be accepted after a minor revision.
Response: The authors thank the Reviewer for the constructive review.
Changes: Changes were made throughout the manuscript according to the suggestions below.
2. line 40: there is the citation [6] without the square brackets.
Response: Thank you. Proofread and corrected.
Changes: Line 40
3. line 42: I suggest using the number "2" instead of "two".
Response: Thank you. Proofread and corrected.
Changes: Line 42.
4. line 72: you wrote "radio therapy". You should cancel the space and write "radiotherapy" (or "Radiation therapy").
Response: Thank you. Proofread and corrected.
Changes: Line 72.
5. line 92: from which molar to which molar the mucoperiosteal incision extends? Regardless instead of "from molar area to molar area" I suggest "from the molar area to the contralateral molar area"
Response: The authors thank the Reviewer’s suggestion. The text was adapted as suggested.
Changes: Lines 93 and 94
6. line 96: in how many cases was the DTX Studio Implant Software used to plan the surgery? Why was the DTX Studio Implant Software used only in certain cases and not in others?
Response: The authors thank the Reviewer’s query. Considering the DTX Studio Implant Software was only used in the last 2 cases as it was only available at that time for use, and that it was not the focus of the study, the authors removed the phrase.
Changes: Lines 97-99.
7. line 300: there is the citation [11] without the square brackets.
Response: Thank you. Proofread and corrected.
Changes: Line 303.
Reviewer 2 Report
Dear Authors,
The article: Zygomatic implants placed in immediate function through extra-maxillary surgical technique and 45 to 60 degrees angulated abutments for full-arch rehabilitation of extremely atrophic maxillae: Short-term outcome of a retrospective cohort was to report the short-term outcome of fixed prosthetic rehabilitations of the atrophic maxillae supported by zygomatic implants placed through the extra-maxillary surgical technique and 45- and 60-degrees angulated abutments.
Please remove academic titles from authors list.
Abstract should be unstructurized.
English language and style are fine.
Correct punctuation mistakes.
Materials and methods:
- please correct bioethical commitee number: Ethical Committee (XXXXXXXXXX, authorization no. 003/2019)
- delete page 5 (it is empty).
Results:
- Table 1 - add abbreviations (sex 1 and 2 female or male)
- Figure 3 and 4 - give better quality of figures.
- line 264 slip to page 11
Add tables with abbreviations used in the article (before references part)
Correct the references using the JCM style.
To sum up, article can be accepted aftrer major revision.
Author Response
Dear Authors,
The article: Zygomatic implants placed in immediate function through extra-maxillary surgical technique and 45 to 60 degrees angulated abutments for full-arch rehabilitation of extremely atrophic maxillae: Short-term outcome of a retrospective cohort was to report the short-term outcome of fixed prosthetic rehabilitations of the atrophic maxillae supported by zygomatic implants placed through the extra-maxillary surgical technique and 45- and 60-degrees angulated abutments.
- Please remove academic titles from authors list.
Response: The authors thank the Reviewer’s indication. The academic titles were removed.
Changes: Lines 6 and 7.
- Abstract should be unstructurized.
Response: The authors thank the Reviewer’s indication. The abstract was adapted as indicated.
Changes: Lines 16 to 30.
- English language and style are fine.
Response: Thank you!
Changes: None.
- Correct punctuation mistakes.
Response: The authors tank the Reviewer’s indication. Proofread and corrected.
Changes: Throughout the manuscript.
Materials and methods:
5. - please correct bioethical committee number: Ethical Committee (XXXXXXXXXX, authorization no. 003/2019)
Response: The authors thank the Reviewer’s indication. The Ethical Committee name (Ethical Committee for Health) was introduced as requested.
Changes: Lines 63,64.
- delete page 5 (it is empty).
Response: Thank you. Proofread and corrected
Changes: Page 5
Results:
7. - Table 1 - add abbreviations (sex 1 and 2 female or male)
Response: The authors thank the Reviewer’s indication. The abbreviations were introduced as requested.
Changes: Table 1, line 215
- Figure 3 and 4 - give better quality of figures.
Response: The authors thank the Reviewer’s indication. The figures were replaced by better quality images.
Changes: Lines 250-261.
- line 264 slip to page 11
Response: The authors thank the Reviewer’s indication. Proofread and corrected.
Changes: line 264
- Add tables with abbreviations used in the article (before references part)
Response: The authors thank the Reviewer’s indication. The indication of name abbreviations used in the article (before references part) were introduced as requested.
Changes: Lines 347-349.
- Correct the references using the JCM style.
Response: The authors thank the Reviewer’s indication. The references were corrected using JCM style.
Changes: Lines 372-439
- To sum up, article can be accepted aftrer major revision.
Response: The authors thank the Reviewer’s constructive review and hope to have complied with all indications.
Round 2
Reviewer 2 Report
Dear Authors,
Thank you for corrections.
Article should be accepted after editors decision.